# Baseline Tyrosine Level Is Associated with Dynamic Changes in FAST Score in NAFLD Patients under Lifestyle Modification

**DOI:** 10.3390/metabo13030444

**Published:** 2023-03-17

**Authors:** Hwi Young Kim, Da Jung Kim, Hye Ah Lee, Joo-Youn Cho, Won Kim

**Affiliations:** 1Department of Internal Medicine, College of Medicine, Ewha Womans University, Seoul 07985, Republic of Korea; 2Metabolomics Core Facility, Department of Transdisciplinary Research and Collaboration, Biomedical Research Institute, Seoul National University Hospital, Seoul 03082, Republic of Korea; 3Clinical Trial Center, Ewha Womans University Medical Center, Seoul 07985, Republic of Korea; 4Department of Clinical Pharmacology and Therapeutics, Seoul National University College of Medicine and Hospital, Seoul 03080, Republic of Korea; 5Department of Biomedical Sciences, Seoul National University College of Medicine, Seoul 03080, Republic of Korea; 6Department of Internal Medicine, Seoul National University College of Medicine and Seoul Metropolitan Government Boramae Medical Center, Seoul 07061, Republic of Korea

**Keywords:** steatosis, steatohepatitis, nonalcoholic steatohepatitis, risk stratification, prediction, multiomics, genomics, metabolomics, weight change, outcome

## Abstract

Noninvasive risk stratification is a challenging issue in the management of patients with nonalcoholic fatty liver disease (NAFLD). This study aimed to identify multiomics-based predictors of NAFLD progression, as assessed by changes in serial FibroScan-aspartate aminotransferase (FAST) scores during lifestyle modification. A total of 266 patients with available metabolomics and genotyping data were included. The follow-up sub-cohort included patients with paired laboratory and transient elastography results (n = 160). The baseline median FAST score was 0.37. The *PNPLA3* rs738409 genotype was significantly associated with a FAST score > 0.35. Circulating metabolomics significantly associated with a FAST score > 0.35 included SM C24:0 (odds ratio [OR] = 0.642; 95% confidence interval [CI], 0.463–0.891), PC ae C40:6 (OR = 0.477; 95% CI, 0.340–0.669), lysoPC a C18:2 (OR = 0.570; 95% CI, 0.417–0.779), and tyrosine (OR = 2.743; 95% CI, 1.875–4.014). A combination of these metabolites and *PNPLA3* genotype yielded a c-index = 0.948 for predicting a FAST score > 0.35. In the follow-up sub-cohort (median follow-up = 23.7 months), 47/76 patients (61.8%) with a baseline FAST score > 0.35 had a follow-up FAST score ≤ 0.35. An improved FAST score at follow-up was significantly associated with age, serum alanine aminotransferase, and tyrosine. In conclusion, baseline risk stratification in NAFLD patients may be assisted using a multiomics-based model. Particularly, patients with increased tyrosine may benefit from an earlier switch to pharmacologic approaches.

## 1. Introduction

The prevalence of nonalcoholic fatty liver disease (NAFLD) is estimated at ~25% worldwide [1]. A subset of these patients will develop nonalcoholic steatohepatitis (NASH), a subset at an elevated risk of disease progression and thereby subject to new pharmacotherapies [2]. Because NASH is estimated to affect up to 20% of individuals with NAFLD, a large number of NASH patients could eventually develop advanced liver disease, or even require liver transplantation [3].

A liver biopsy has been the gold standard for the diagnosis of NASH [4]. However, the need for alternative noninvasive tests or biomarkers has been growing owing to the imperfect nature of biopsy, which considers the invasiveness, risk of complications, cost, sampling error, and inter-/intra-observer variability [5]. Furthermore, identification of patients with high-risk NASH (NAFLD activity score ≥ 4 and significant [≥F2] liver fibrosis) has been a pressing issue in clinical trial eligibility because those patients are considered to be at greatest risk of disease progression and liver-related morbidity and mortality [2,6]. In that context, the FibroScan-aspartate aminotransferase (FAST) score has been developed recently for noninvasive identification of high-risk NASH based on aspartate aminotransferase (AST), liver stiffness measurement (LSM), and a controlled attenuation parameter (CAP) [7]. In addition, multiomics approaches have been widely investigated for genotype–phenotype correlation, the development of biomarkers, and the identification of therapeutic targets [8].

Although there are various ongoing clinical trials, no pharmacologic agent has been approved for NASH yet, and lifestyle modification is the mainstay of management for most patients with NAFLD [9]. To date, there are limited data on integrative methods for a noninvasive risk stratification of NAFLD and early identification of individuals who may benefit from lifestyle modification. Filling in these knowledge gaps could provide tailored practical information on noninvasive risk stratification of NAFLD and early application of upcoming pharmacotherapies. Hence, we aimed to explore multiomics-based predictors of baseline risk stratification and NAFLD progression based on changes in the serial FAST scores during lifestyle modification in a real-world setting.

## 2. Methods

### 2.1. Study Participants

We constructed a single-center prospective cohort of Korean patients with NAFLD who had been referred to our liver clinic since October 2016 (see Appendix A). Patients were either referred by their primary care providers or referred from interdepartmental consultations inside our institution. The following exclusion criteria were applied: (i) age <18 years, (ii) hepatitis B or C virus infection, (iii) presence of other chronic liver diseases (e.g., autoimmune hepatitis, primary biliary cholangitis or primary sclerosing cholangitis, drug-induced liver injury or steatosis, Wilson’s disease, and hemochromatosis), (iv) excessive alcohol consumption (>30 g/day in men and >20 g/day in women) [4], and (v) diagnosis of malignancy.

This was a retrospective analysis from our prospectively enrolled cohort. A total of 423 NAFLD patients were enrolled between October 2016 and December 2020. Of these, 266 patients with available metabolomics and genotyping data were included in the cross-sectional analysis. All study participants were consulted for dietary and exercise education at baseline according to practice guidelines (see Appendix A). A subset of patients was identified with one or more follow-up visit(s) until the closure date for data analysis (31 March 2022), and those with paired laboratory and vibration-controlled transient elastography (VCTE) results (n = 160, “follow-up sub-cohort”) were included in the longitudinal analysis.

The present study was conducted in accordance with the ethical guidelines of the World Medical Association’s Declaration of Helsinki and was approved by the institutional review board of Ewha Womans University Mokdong Hospital (approval no.: EUMC 2016–07–052; approval date: 8-30-2016). Written informed consent was obtained from each participant in the cohort.

### 2.2. Baseline Clinical and Laboratory Assessments

Initial assessments included anthropometric measurements, routine laboratory tests, body composition analysis, and VCTE. Body composition was assessed using bioelectrical impedance analysis (InBody 720 body composition analyzer, InBody, Seoul, Korea) (see Appendix A). VCTE procedures were performed after fasting for at least 3 h using the FibroScan 502 Touch device equipped with both M and XL probes (Echosens, Paris, France). Study participants were placed in the supine position with their right arm fully abducted. Upon obtaining 10 valid measurements, final values of CAP (dB/m) and LSM (kPa) were recorded as the median values of 10 consecutive valid measurements. LSM values were considered unreliable when the interquartile range (IQR)/median was higher than 30% [10]. All VCTE procedures were performed by research personnel who were trained and certified by Echosens and who were blinded to the clinical and laboratory details of the participants at the time of the examination.

### 2.3. Metabolomics and Genotyping

Targeted metabolomics were assessed using a Triple Quadrupole 6500 plus system (AB Sciex, Framingham, MA, USA), which consists of a SHIMADZU Nexera (Shimadzu Corporation, Kyoto, Japan) ultra-high performance liquid chromatography coupled with a hybrid triple quadrupole/linear ion trap mass spectrometer. A total of 180 metabolites were quantified using an Absolute IDQ^®^p180 kit (BIOCRATES Life Science AG, Innsbruck, Austria). The kit allowed the concurrent high-throughput detection and quantification of metabolites in plasma samples. Genotyping was performed for several known risk alleles for NAFLD as follows: *PNPLA3* rs738409 C>G, *TM6SF2* rs58542926 C>T, *SREBF2* rs133291 C>T, *MBOAT7-TMC4* rs641738 C>T, and *HSD17B13* rs72613567 adenine insertion (A-INS) single-nucleotide polymorphisms (see Appendix A).

### 2.4. Statistical Analysis

The statistical significance of differences between groups was evaluated using the independent *t*-test or Mann–Whitney U test for continuous variables and the chi-square test for categorical variables. Relevant risk factors for the outcome were explored with logistic regression analysis using baseline clinical characteristics, metabolomics, and genotyping data (see Appendix A). To identify significantly different metabolites, *p*-values were adjusted for multiple testing using the Benjamini–Hochberg procedure for conceptualizing the false discovery rate (FDR). Multiple logistic regression was used to investigate the independent factors determining the risk groups according to the FAST score. A multivariable model was constructed through stepwise selection among candidate risk factors with *p* < 0.05 in the univariable analysis. Model performance was presented using the concordance index (c-index) and their 95% confidence intervals were estimated using 1000 bootstrap samples. Hosmer–Lemeshow goodness-of-fit tests were conducted as calibration statistics.

For the follow-up sub-cohort, the mean difference in changes in clinical parameters according to the level of weight change during the follow-up period was tested using an analysis of covariance (ANCOVA). Additionally, among patients with baseline a FAST score > 0.35, a multiple logistic regression model and mediation analysis were performed to evaluate the determining factors associated with a categorical change in the FAST score to a low risk (≤0.35). ANCOVA and mediation analysis were assessed by adjusting for sex, age, follow-up duration, baseline weight, and baseline clinical parameters.

All statistical tests were two-sided with *p* < 0.05 as the threshold for statistical significance. The SAS 9.4 (SAS Institute, Cary, NC, USA) and R 3.6.2 software packages (R Foundation for Statistical Computing, Vienna, Austria) were used for all statistical analyses.

## 3. Results

### 3.1. Baseline Characteristics

The baseline FAST score was 0.37 (IQR, 0.14–0.53). Compared with patients with a lower FAST score (≤ 0.35, n = 126 [47.4%]), patients with a higher score (>0.35, n = 140 [52.6%]) showed higher BMI, waist circumference, glycometabolic parameters, liver injury markers, CAP (305 vs. 271 dB/m), and LSM (7.7 vs. 4.6 kPa), and more frequently had metabolic syndrome (65.0% vs. 42.9%) and sarcopenia (38.6% vs. 20.7%); all *p*-values <0.05 (Table 1). There was no significant difference in the minor allele frequency of each genotype between the two groups (FAST score ≤ 0.35 vs. > 0.35; Table 1).

### 3.2. Risk Factors for Higher FAST Score (>0.35) at Baseline

In univariable analysis, significant risk factors for a higher FAST score (>0.35) included the following (Table 2): higher values of BMI, waist circumference, ALT, GGT, fasting blood glucose, insulin, WBC, HOMA-IR, and fat%; lower values of HDL-cholesterol and SMI_wt; and the presence of metabolic syndrome and diabetes. Among genotypes, only *PNPLA3* rs738409 was significantly associated with a higher FAST score.

In Table 3, circulating metabolites had significant correlations with a higher FAST score (>0.35), as seen with sphingomyelin (SM [OH] C22:2, odds ratio [OR] = 0.53, 95% confidence interval [CI] 0.41–0.70, *P_FDR_* = 0.001; SM C24:0, OR = 0.59, 95% CI 0.45–0.77, *P_FDR_* = 0.003; and SM C16:0, OR = 0.59, 95% CI, 0.45–0.77, *P_FDR_* = 0.004), phosphatidyl choline (PC ae C40:6, OR = 0.56, 95% CI 0.43–0.74, *P_FDR_* = 0.003; PC ae C38:0, OR = 0.58, 95% CI 0.44–0.76, *P_FDR_* = 0.004; and lysoPC a C18:2, OR = 0.61, 95% CI 0.47–0.80, *P_FDR_* = 0.004), and tyrosine (OR = 1.56, 95% CI 1.19–2.03, *P_FDR_* = 0.013).

In multiple logistic regression analysis (Table 4), relevant risk factors for a higher FAST score were identified as follows: age, ALT, HOMA-IR, sarcopenia, *PNPLA3* genotype (dominant model), PC ae C40:6, lysoPC a C18:2, SM C24:0, and tyrosine. The multiomics-based prediction model (Model 3, c-index = 0.948; 95% CI 0.927–0.978; *p* for Hosmer–Lemeshow = 0.190) and the clinico-genomic model (Model 2, c-index = 0.933; 95% CI 0.906–0.964; *P* for Hosmer–Lemeshow = 0.209) yielded a higher predictive performance compared to the metabolomics-based model (Model 1, c-index = 0.782; 95% CI 0.734–0.838; *P* for Hosmer–Lemeshow = 0.246) (*p* < 0.001).

### 3.3. Follow-Up Sub-Cohort (n = 160)

A total of 160 patients who were followed up more than twice with at least one follow-up VCTE during the study period comprised the follow-up sub-cohort. During a median follow-up of 23.7 months (IQR, 13.2–33.8), weight loss > 5% between baseline and the last visit was observed in 30 patients (18.7%), weight loss ≤ 5% was observed in 75 patients (46.9%), and weight gain was observed in 55 patients (34.4%). Figure 1 depicts the changes in the FAST score (≤ 0.35 vs. 0.35–0.67 vs. ≥ 0.67) according to the level of weight change under lifestyle modification. Low-risk patients at baseline (FAST ≤ 0.35, n = 84) mostly remained in the same category at follow-up (79, 94.0%). However, 47 out of 76 patients (61.8%) with a baseline FAST score > 0.35 were classified as low-risk NASH (FAST score ≤ 0.35) at follow-up (Appendix A).

Significant differences in terms of changes in skeletal muscle mass, fat%, ALT, GGT, FAST score, and LSM were observed among the three subgroups according to the extent of weight change (all *p* < 0.05; Appendix A). Multiple logistic regression analysis of the follow-up sub-cohort showed that age, ALT, and tyrosine were significantly associated with a shift in the FAST score toward a low-risk score (≤0.35) from a baseline level of >0.35, regardless of baseline weight or weight changes during follow-up (Table 5).

In the mediation analysis, in patients with a baseline FAST score > 0.35, weight gain was associated with an increase in serum ALT (*β* = 3.154, 95% CI 1.283–5.026, *p* = 0.001). A follow-up FAST score ≤ 0.35 showed a significant inverse relationship with an increase in serum ALT (*β* = −0.083, 95% CI −0.127–−0.039, *p* < 0.001). The indirect effect of ALT as a mediator on the association between changes in weight and a shift toward a FAST score ≤ 0.35 was statistically significant (Appendix A).

## 4. Discussion

The present exploratory study demonstrated that the combination of clinical (age, ALT, HOMA-IR, and sarcopenia), metabolomic (sphingolipids, glycerophospholipids, and tyrosine), and genomic (*PNPLA3* genotype) biomarkers accurately predicted a higher FAST score (>0.35). In the follow-up sub-cohort, predictors of a shift of the FAST score toward a low-risk score (≤0.35) following lifestyle modification included a younger age and lower baseline levels of serum ALT and tyrosine. Mediation analysis suggested that a downward shift in the FAST score in association with weight change might be mediated by the mitigation of hepatic inflammation.

Identification of patients at high risk of disease progression is a challenging issue [2]. Given the inherent limitations of liver biopsy, noninvasive tests (NITs) have been developed for routine practice and sustainable clinical care pathways. However, longitudinal changes in those NITs following certain treatments remain to be elucidated because those NITs were derived and validated in cross-sectional settings [11]. In the current study, at baseline, 47.4% of participants were classified as low-risk (FAST ≤ 0.35), 10.1% of participants were classified as high-risk (FAST ≥ 0.67), and 42.5% of participants were classified as in the gray zone (0.35 < FAST < 0.67). The FAST score demonstrated a good diagnostic performance for high-risk NASH in the derivation cohort as well as in the external validation cohorts [7]. However, an independent external validation study by Puri et al. involving 199 United States veterans reported a lower positive predictive value (0.26) of an upper cut-off (0.67) to rule in, compared with the original study, along with a high proportion of the gray zone (35.5%) [12]. Given that the diagnostic performance of biomarkers depends on different clinical settings, the major implication of the FAST score in the present study could be the confirmatory exclusion of patients with high-risk or fibrotic NASH based on the lower cut-off [13]. Another concern in relation to the FAST score seems to be its inherently high dependency on AST levels, considering that most patients with advanced liver fibrosis on biopsy present with normal aminotransferase levels [14]. Thus, experts recommend paying attention to the LSM per se to minimize the risk of the underestimation of patients with advanced fibrosis, suggesting sequential testing in case of high LSM despite being below a low-risk cut-off [15]. Given the low baseline LSM value (median, 4.6 kPa; IQR, 3.8–6.1) in patients with a FAST score ≤ 0.35 in our real-life cohort (Table 1), the utilization of a lower cut-off (0.35) as an exclusion of high-risk NASH seems justifiable. Thus, we dichotomized study participants according to the lower cut-off (0.35) to distinguish probable low-risk patients from patients falling in the gray zone or high-risk NASH patients who would benefit from further testing or enrollment in clinical trials. In addition, since most low-risk patients remained low risk at follow-up (94%) in the follow-up sub-cohort, we attempted to identify the predictors of a downward shift in the risk group, in which patients in the gray zone and high-risk patients became the low-risk group after follow-up.

In the present study, univariable analysis showed that only the *PNPLA3* rs738409 genotype was significantly associated with a FAST score > 0.35. Among the 34 metabolites which met the FDR-adjusted threshold for a significant association with a higher FAST score, the top four metabolites panel based on the stepwise regression included sphingolipids, glycerophospholipids, and amino acids. A model combining clinical (age, sex, ALT, HOMA-IR, and sarcopenia), genomic (*PNPLA3* rs738409), and metabolomic (PC ae C40:6, lysoPC a C18:2, SM C24:0, and tyrosine) parameters achieved the highest c-index (0.948) for the prediction of high-risk NASH. Low concentrations of certain glycerophospholipids (acyl-alkyl PC or PC ae) and sphingolipids were characteristic of NAFLD in previous studies, suggesting the increased turnover and size of adipocytes, which necessitate high PC amounts for cell membrane production in the pathogenesis of NAFLD [16]. Sphingolipids and PCs are both biochemically related phospholipids and key components of the cell membrane, which explains a significant inverse relationship with a higher FAST score (>0.35) or low likelihood of high-risk NASH, as our results indicated. In addition, lysophosphatidylcholines (lysoPCs) were associated with liver fat content as the hallmark of NAFLD in a Finnish study [17]. Subsequent evidence was added regarding lysoPC a C18:2 as a marker of impaired glucose tolerance [18], obesity [19], and type 2 diabetes [20]. In our results, lysoPC a C18:2 was inversely associated with a FAST score > 0.35. Given that lysoPCs are formed by the oxidation of PCs in phospholipid-containing cell membranes or low-density lipoprotein (LDL) [21], this inverse relationship might result from an increased breakdown of lipid from metabolically active tissues, which is in line with previous studies [18,20].

Alterations in circulating amino acids have been documented in patients with NAFLD, including increases in branched chain amino acids or aromatic amino acids [22]. The altered hepatic amino acid composition implies the role of amino acids as adaptive response mechanisms to lipotoxicity in progressive NAFLD [23]. Specifically, tyrosine has been reported to be a marker of development of insulin resistance, upregulated in NAFLD/NASH patients, and positively correlated with total and LDL cholesterol [24,25,26]. Decreased gene expression of amino acid transporters, such as SLC16A10, was suggested to be a potential mechanism for the elevation of tyrosine levels in NASH patients [24], which could not be verified due to lack of sufficient liver samples. Although the mechanism of tyrosine dysregulation in NAFLD remains to be elucidated, the blood tyrosine level has been arguably identified as a potential biomarker for NAFLD [27], which might also be the marker of risk stratification for fibrotic NASH, in agreement with our results. Furthermore, tyrosine was the only significant metabolite in the prediction of a FAST score ≤ 0.35 at follow-up in patients with a baseline FAST score > 0.35. A recent Finnish study suggested that an exercise-related decrease in fasting plasma glucose and improved cardiorespiratory fitness might be derived by the increased tyrosine level in the adipose tissue, given tyrosine is a catecholamine precursor which plays a vital role in athletic performance and lipolysis [28]. Our results might also reflect the similar role of tyrosine in patients undergoing lifestyle modification, which requires further validation. Lastly, in the mediation analysis, the significant indirect effect of serum ALT on weight changes and outcomes (e.g., FAST score at follow-up) is in line with previous reports in which beneficial effects of lifestyle modification result from the amelioration of hepatic inflammation with or without an improvement in fibrosis according to the extent of weight loss [29]. However, ALT change was the only statistically significant mediator in the relationship between weight change and the follow-up FAST score in the mediation analysis. This finding suggests that the mediation analysis was possibly not able to capture certain beneficial effects of lifestyle modification on other pathophysiologic aspects of NAFLD/NASH, such as metabolic comorbidities. One possible explanation would be the insufficient follow-up duration (median 23.7 months) and/or number of study participants. In addition, sarcopenia was not significantly associated with the outcome, unlike previous reports, possibly because of the relatively low proportion of high-risk (FAST ≥ 0.67) patients (10.1%) [30,31].

This study has several limitations. First, and most importantly, a liver biopsy was performed in only a subset of patients, which raises the possibility of misclassification bias and precluded an external validation of the FAST score or comparison with other noninvasive tests, using histologic data as a gold standard. However, performing a biopsy in the entire study participants would be unethical given the substantial proportion of the low-risk group (FAST ≤ 0.35). Instead, we focused on the following questions: who would be in the low-risk group and who would not, given that there are no widely approved pharmacological agents against NASH and virtually all patients with NAFLD should undergo lifestyle modification. In that context, the dichotomous categorization into low-risk vs. gray zone + high-risk according to the lower cut-off (0.35) seems reasonable considering the low LSM values in the low-risk group, instead of using three categories without reference to the histological data. Second, 39.9% (106/266) of the entire cohort were lost to the follow-up, which might have affected the results of the present study by way of the Hawthorne effect [32]. In addition, the relatively short duration of the follow-up (2 years) might have affected the lower frequency of progression in the FAST score in the follow-up sub-cohort. However, lifestyle modification was presumed to be implemented strictly, given that the low-risk FAST score was maintained in 94% of the follow-up sub-cohort, which reduces concerns in relation to compliance bias. Third, the observational nature of this study might preclude the assessment of causality, mechanistic links, and roles of the identified risk factors.

In conclusion, the present study suggests the potential usefulness of a multiomics-based risk categorization based on glycerophospholipids (PC ae C40:6, lysoPC a C18:2), sphingolipids (SM C24:0), and amino acids (tyrosine) at baseline as well as at follow-up after lifestyle modification. Despite the aforementioned limitations, our results provide solid data on the dynamic change of paired NITs and would facilitate the development of a risk stratification strategy before and after lifestyle modification if properly validated.

## Figures and Tables

**Figure 1 metabolites-13-00444-f001:**
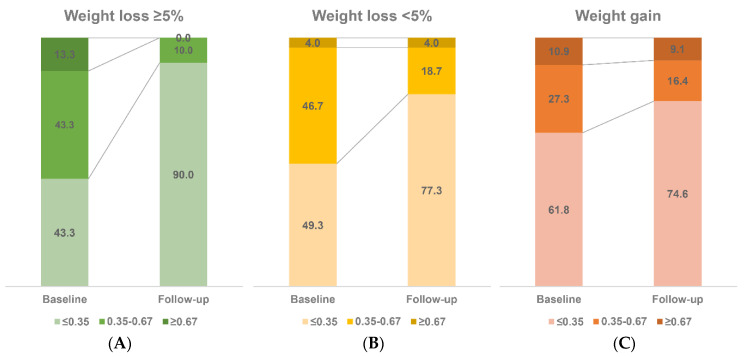
Changes in the FAST score during follow-up. Distribution of changes in the FAST score between baseline and the last follow-up, according to the levels of weight change under lifestyle modification (≥5% weight loss (**A**), <5% weight loss (**B**), and weight gain (**C**)).

**Table 1 metabolites-13-00444-t001:** Baseline characteristics.

		*Total (* *N* *= 266)*	*FAST* *≤* *0.35* *(n = 126, 47.4%)*	*FAST > 0.35* *(n = 140, 52.6%)*	*p*
Age		49.6 ± 14.3	50.3 ± 12.7	49.1 ± 15.7	0.478
Sex (male)		158 (59.4%)	77 (61.11%)	81 (57.8%)	0.589
BMI (kg/m^2^)		27.2 ± 3.6	26.1 ± 3.1	28.3 ± 3.8	<0.001
Waist circumference (cm)		93.4 ± 9.9	91.5 ± 8.9	95.4 ± 10.5	0.002
Metabolic syndrome		145 (54.5%)	54 (42.9%)	91 (65.0%)	0.001
Hypertension		83 (31.2%)	33 (26.19%)	50 (35.71%)	0.094
Diabetes		61 (22.93%)	17 (13.39%)	44 (31.43%)	0.001
Dyslipidemia		37 (13.9%)	21 (16.67%)	16 (11.43%)	0.228
AST (IU/L)		38 (26–60)	26 (21–32)	56.5 (43.5–76)	<0.001
ALT (IU/L)		52 (31–94)	31 (20–48)	88 (57–120.5)	<0.001
GGT (IU/L)		44 (27–72)	32 (22–52)	59 (36–89)	<0.001
Glucose (mg/dL)		103 (96–115)	100 (95–108)	105.5 (97.5–123)	0.002
Cholesterol (mg/dL)		195.8 ± 45.5	200 ± 47.97	191.99 ± 42.87	0.149
TG (mg/dL)		142.5 (100–201)	129 (84–186)	152 (113–213.5)	0.005
HDL (mg/dL)		47.2 ± 12.2	49.7 ± 13.2	44.9 ± 10.8	0.002
LDL (mg/dL)		122.7 ± 40.2	124.5 ± 43.4	121.0 ± 37.1	0.484
Insulin (μIU/mL)		11.9 (7.9–18.9)	9.1 (6.4–13.8)	15.3 (10.4–25.8)	<0.001
Uric acid (mg/dL)		5.9 ± 1.5	5.8 ± 1.4	6.0 ± 1.6	0.261
FFA (mmol/L)		819.1 ± 314.5	774.8 ± 339.3	858.8 ± 286.1	0.048
WBC (10^3^/µL)		6.73 ± 1.72	6.37 ± 1.68	7.05 ± 1.70	0.002
Hemoglobin (g/dL)		14.7 ± 1.6	14.7 ± 1.5	14.7 ± 1.7	0.981
Platelet (10^3^/µL)		244.3 ± 64.0	245.8 ± 55.8	244.9 ± 69.1	0.910
Serum creatinine (mg/dL)		0.91 ± 0.18	0.92 ± 0.19	0.90 ± 0.17	0.237
HOMA-IR		3.2 (2.0–5.4)	2.3 (1.6–3.6)	4.2 (2.8–7.2)	<0.001
SMI_wt		28.0 ± 3.9	28.8 ± 3.6	27.2 ± 4.0	0.001
Sarcopenia		74 (29.8%)	25 (20.7%)	49 (38.6%)	0.002
Fat%		32.4 ± 7.9	30.6 ± 7.7	34.1 ± 7.7	<0.001
Handgrip strength (kg)		34.8 ± 10.9	35.1 ± 10.7	34.4 ± 11.1	0.688
TSH (mIu/L)		2.1 (1.3–3.1)	2.1 (1.4–3.2)	2.1 (1.3–3.1)	0.695
Free T4 (ng/dL)		1.3 (1.2–1.4)	1.3 (1.2–1.4)	1.3 (1.1–1.4)	0.093
HbA1c		5.9 (5.5–6.6)	5.8 (5.3–6.2)	6.0 (5.6–6.9)	0.009
CAP (dB/min)		289.3 ± 43.5	271.4 ± 39.6	305.4 ± 40.6	<0.001
LSM (kPa)		6.2 (4.6–8.6)	4.6 (3.8–6.1)	7.7 (6.1–10.2)	<0.001
FAST score		0.37 (0.14–0.53)	0.14 (0.09–0.25)	0.52 (0.43–0.64)	<0.001
PNPLA3 rs738409	C/C	61 (23.3%)	36 (28.8%)	25 (18.3%)	0.129
	C/G	130 (49.6%)	58 (46.4%)	72 (52.5%)	
	G/G	71 (27.1%)	31 (24.8%)	40 (29.2%)	
TM6SF2 rs58542926	C/C	222 (84.7%)	108 (86.4%)	114 (83.2%)	0.534
	C/T	39 (14.9%)	17 (13.6%)	22 (16.1%)	
	T/T	1 (0.4%)	0 (0%)	1 (0.7%)	
MBOAT7 rs641738	C/C	154 (59.3%)	75 (60.5%)	79 (58.1%)	0.729
	C/T	95 (36.5%)	45 (36.3%)	50 (36.8%)	
	T/T	11 (4.2%)	4 (3.2%)	7 (5.1%)	
SREBF2 rs133291	C/C	84 (32.1%)	39 (31.2%)	45 (32.9%)	0.950
	C/T	142 (54.2%)	69 (55.2%)	73 (53.3%)	
	T/T	36 (13.7%)	17 (13.6%)	19 (13.9%)	
HSD17B13 rs72613567	−/−	137 (52.3%)	69 (55.2%)	68 (49.64%)	0.665
	−/A	103 (39.3%)	46 (36.8%)	57 (41.61%)	
	A/A	22 (8.4%)	10 (8%)	12 (8.76%)	

The continuous variables are expressed as the means ± standard deviations (normally distributed) or medians (interquartile ranges) (not normally distributed), and the differences between groups were evaluated using an independent *t*-test or Mann–Whitney U test, respectively. Categorical data were expressed as the number (%), and the differences between groups were determined using the χ2 test. Abbreviations: BMI, body mass index; AST, aspartate aminotransferase; ALT, alanine aminotransferase; GGT, γ-glutamyl transpeptidase; TG, triglyceride; HDL, high-density lipoprotein; LDL, low-density lipoprotein; FFA, free fatty acid; WBC, white blood cell; HOMA-IR, homeostasis model assessment of insulin resistance; SMI_wt, weight-adjusted skeletal muscle index; fat%, fat percentage; TSH, thyroid stimulating hormone; HbA1c, glycated hemoglobin; CAP, controlled attenuated parameter; LSM, liver stiffness measurement; FAST, FibroScan-aspartate aminotransferase.

**Table 2 metabolites-13-00444-t002:** Clinical characteristics and genetic risk factors associated with high-risk NASH (FAST score > 0.35).

*Variable*	*OR*	*95% CI*	*p*
Age	0.99	0.98–1.01	0.482
Sex (male)	1.14	0.70–1.87	0.590
BMI	1.20	1.11–1.30	<0.001
Waist circumference	1.04	1.02–1.07	0.003
Metabolic syndrome	1.57	1.27–1.93	<0.001
Hypertension	1.57	0.93–2.65	0.095
Diabetes	2.94	1.58–5.48	0.001
Dyslipidemia	0.65	0.32–1.30	0.220
ALT	1.06	1.05–1.08	<0.001
GGT	1.02	1.01–1.03	<0.001
Glucose	1.02	1.01–1.04	0.002
Cholesterol	1.00	0.99–1.00	0.151
TG	1.00	1.00–1.00	0.101
HDL	0.97	0.95–0.99	0.002
LDL	1.00	0.99–1.00	0.483
Insulin	1.10	1.06–1.14	<0.001
WBC	1.28	1.09–1.50	0.003
HOMA-IR	1.45	1.27–1.66	<0.001
SMI_wt	0.90	0.84–0.96	0.001
Fat%	1.06	1.03–1.10	0.001
PNPLA3 (ref. C/C)			
C/G	2.02	1.09–3.88	0.034
G/G	2.10	1.01–4.37	0.047
linear (per 1 risk allele)	1.35	0.96–1.91	0.089
C/G+G/G vs. C/C	1.81	1.01–3.24	0.045
TM6SF2 (ref. C/C)			
C/T	1.30	0.65–2.60	0.464
linear (per 1 risk allele)	1.33	0.69–2.56	0.398
C/T, T/T vs. C/C	1.28	0.65–2.53	0.474
MBOAT7 (ref. C/C)			
C/T	1.06	0.63–1.76	0.838
T/T	1.66	0.47–5.91	0.433
linear (per 1 risk allele)	1.14	0.75–1.74	0.546
C/T, T/T vs. C/C	1.10	0.67–1.81	0.695
SREBF2 (ref. C/C)			
C/T	0.89	0.51–1.56	0.695
T/T	1.06	0.48–2.34	0.879
linear (per 1 risk allele)	0.97	0.67–1.40	0.864
C/T, T/T vs. C/C	0.93	0.55–1.56	0.776
HSD17B13 (ref. −/−)			
Heterozygous −/A	1.26	0.75–2.10	0.382
Homozygous A/A	1.22	0.49–3.01	0.669

Abbreviations: NASH, nonalcoholic steatohepatitis; FAST, FibroScan-aspartate aminotransferase; BMI, body mass index; ALT, alanine aminotransferase; GGT, γ-glutamyl transpeptidase; TG, triglyceride; HDL, high-density lipoprotein; LDL, low-density lipoprotein; WBC, white blood cell; HOMA-IR, homeostasis model assessment of insulin resistance; SMI_wt, weight-adjusted skeletal muscle index; fat%, fat percentage.

**Table 3 metabolites-13-00444-t003:** Metabolites that were significantly associated with high-risk NASH (FAST score > 0.35).

*Variable*	*OR*	*95% CI*	*Raw* *p–* *Value*	*Rank*	*BH Adjusted* *p–* *Value*
SM (OH) C22:2	0.53	0.41–0.709	5.76193E–06	1	0.001198481
SM (OH) C16:1	0.57	0.43–0.74	3.95722E-05	2	0.004115505
PC ae C40:6	0.56	0.43–0.74	4.39683E-05	3	0.003048471
SM C16:0	0.59	0.45–0.77	7.63941E-05	4	0.003972496
PC ae C38:0	0.58	0.44–0.76	9.78989E-05	5	0.004072592
SM C24:0	0.59	0.45–0.77	9.88366E-05	6	0.003426335
PC ae C40:5	0.57	0.43–0.76	0.000114559	7	0.003404043
PC aa C38:0	0.60	0.46–0.78	0.000149	8	0.003874005
SM C24:1	0.60	0.46–0.78	0.000155451	9	0.003592655
SM (OH) C22:1	0.60	0.46–0.79	0.000179669	10	0.003737106
SM C16:1	0.61	0.47–0.79	0.000194861	11	0.003684646
PC ae C38:6	0.61	0.47–0.79	0.000200649	12	0.003477909
PC ae C36:2	0.61	0.47–0.79	0.000220253	13	0.003524049
LysoPC a C18:2	0.61	0.47–0.80	0.000289073	14	0.004294803
PC aa C36:6	0.62	0.48–0.81	0.000311757	15	0.004323034
PC ae C40:4	0.60	0.46–0.80	0.000394197	16	0.005124564
PC aa C36:5	0.62	0.48–0.82	0.000557316	17	0.006818923
Tyrosine	1.56	1.19–2.03	0.00108511	18	0.012539053
PC aa C42:4	1.99	1.30–3.04	0.001440041	19	0.01576466
PC ae C38:5	0.66	0.51–0.86	0.001646602	20	0.017124665
DCA	1.57	1.18–2.08	0.001794496	21	0.017774053
GLCA	2.06	1.30–3.27	0.002110225	22	0.019951219
LCA	1.89	1.26–2.83	0.002162058	23	0.019552522
PC ae C36:1	0.68	0.52–0.87	0.002786416	24	0.024148943
SM (OH) C14:1	0.68	0.53–0.88	0.003112874	25	0.025899116
TUDCA	3.10	1.46–6.58	0.003205053	26	0.025640423
PC ae C36:5	0.68	0.53–0.88	0.003436294	27	0.026472192
LysoPC a C18:1	0.68	0.52–0.89	0.004346437	28	0.032287817
GCDCA	1.59	1.16–2.19	0.0044439	29	0.031873489
SM C18:1	0.69	0.54–0.99	0.004463313	30	0.030945637
PC aa C36:0	0.70	0.54–0.90	0.005357294	31	0.035945714
SM C18:0	0.70	0.54–0.90	0.006127907	32	0.039831397
GDCA	1.59	1.14–2.22	0.006191177	33	0.039023178
LysoPC a C17:0	0.69	0.52–0.90	0.00685643	34	0.041945216

Abbreviations: NASH, nonalcoholic steatohepatitis; FAST, FibroScan-aspartate aminotransferase; OR, odds ratio; CI, confidence interval; BH, Benjamini–Hochberg; SM, sphingomyeline; PC, phosphatidylcholine; lysoPC, lysophosphatidylcholine; DCA, deoxycholic acid; GLCA, glyco-lithocholic acid; LCA, lithocholic acid; TUDCA, tauro-ursodeoxycholic acid; GCDCA, glyco-chenodeoxycholic acid; GDCA, glyco-deoxycholic acid.

**Table 4 metabolites-13-00444-t004:** Multiple logistic regression analysis on risk factors for high-risk NASH (FAST score > 0.35).

		*Model 1*			*Model 2*			*Model 3*	
	OR	95% CI	*p*	OR	95% CI	p	OR	95% CI	*p*
PC ae C40:6	0.48	0.34–0.67	<0.001				0.61	0.35–1.04	0.071
lysoPC a C18:2	0.57	0.42–0.78	<0.001				0.72	0.43–1.20	0.201
SM C24:0	0.64	0.46–0.89	0.008				0.57	0.35–0.92	0.022
Tyrosine	2.74	1.88–4.01	<0.001				2.07	1.14–3.78	0.018
Sex				1.03	0.44–2.43	0.945	1.30	0.50–3.39	0.591
Age				1.05	1.02–1.09	0.002	1.04	1.00–1.08	0.035
ALT				1.07	1.05–1.10	<0.001	1.07	1.05–1.09	<0.001
HOMA-IR				2.94	1.20–7.21	0.019	2.14	0.84–5.49	0.113
Sarcopenia				3.14	1.30–7.59	0.011	3.85	1.45–10.26	0.007
PNPLA3				1.53	0.59–3.94	0.384	1.83	0.62–5.40	0.272

Note. Model 1 used metabolomics data; Model 2 used clinical + genomics data; Model 3 used clinical + metabolomics + genomics data. Abbreviations: NASH, nonalcoholic steatohepatitis; FAST, FibroScan-aspartate aminotransferase; OR, odds ratio; CI, confidence interval; SM, sphingomyeline; PC, phosphatidylcholine; lysoPC, lysophosphatidylcholine; ALT, alanine aminotransferase; HOMA-IR, homeostasis model assessment of insulin resistance.

**Table 5 metabolites-13-00444-t005:** Multiple logistic regression analysis on the predictors of NASH resolution defined as follow-up FAST score ≤ 0.35 following lifestyle modification in patients with baseline FAST score > 0.35.

		*Model 1*			*Model 2*	
	OR	95% CI	*p*	OR	95% CI	*p*
PC ae C40:6	2.54	0.91–7.07	0.074	2.47	0.81–7.53	0.111
LysoPC a C18:2	0.97	0.40–2.36	0.940	0.98	0.38–2.54	0.973
SM C24:0	1.10	0.44–2.79	0.835	1.25	0.41–3.80	0.693
Tyrosine	0.36	0.15–0.88	0.025	0.36	0.13–0.97	0.044
Sex	0.86	0.20–3.76	0.836	0.31	0.05–2.07	0.227
Age	0.92	0.86–0.98	0.010	0.88	0.81–0.97	0.007
ALT	0.97	0.95–0.99	0.001	0.96	0.94–0.99	0.002
HOMA-IR	0.84	0.11–6.55	0.867	1.26	0.13–11.99	0.841
Sarcopenia	0.42	0.10–1.84	0.247	0.90	0.13–6.23	0.915
PNPLA3	0.93	0.13–6.65	0.938	0.98	0.11–8.61	0.983
Baseline weight				0.93	0.86–1.01	0.099
Weight change (ref.: weight loss <5%)						
weight loss ≥ 5%				6.25	0.55–70.64	0.138
weight gain				0.65	0.12–3.46	0.612
c–index	0.845 (95% CI 0.815-0.989) *			0.861 (95% CI 0.858-1.000) *		

Note. Model 1 used baseline clinical characteristics and multiomics data; Model 2: Model 1 + weight change. *P* values for Hosmer–Lemeshow test were 0.207 for Model 1, and 0.783 for Model 2, respectively. * Using bootstrap resampling (times = 1000). Abbreviations: NASH, nonalcoholic steatohepatitis; FAST, FibroScan-aspartate aminotransferase; OR, odds ratio; CI, confidence interval; SM, sphingomyeline; PC, phosphatidylcholine; lysoPC, lysophosphatidylcholine; ALT, alanine aminotransferase; HOMA-IR, homeostasis model assessment of insulin resistance.

## Data Availability

The data presented in this study are available on reasonable request from the corresponding author. Data is not publicly available due to privacy.

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
