# Peer review of "Baseline Tyrosine Level Is Associated with Dynamic Changes in FAST Score in NAFLD Patients under Lifestyle Modification"

_metabolites, 2023, doi:10.3390/metabo13030444_

Round 1

Reviewer 1 Report

In this study, the authors attempted to identify biomarker for NAFLD by determining the changes in the FibroScan-aspartate aminotransferase scores in 266 NAFLD patients. The study is interesting and there are few concerns the authors need to address

1.       Did the authors compare FAST score with the histological data of NAFLD patients?

2.       Did the authors find similar changes among other aromatic amino acids and FAST score like that of tyrosine?

Author Response

In this study, the authors attempted to identify biomarker for NAFLD by determining the changes in the FibroScan-aspartate aminotransferase scores in 266 NAFLD patients. The study is interesting and there are few concerns the authors need to address

  1. Did the authors compare FAST score with the histological data of NAFLD patients?

Response: We thank the reviewer for this important comment. Due to the small proportion of patients who underwent biopsy (n=41, 15.4%), we did not obtain meaningful results on the comparison between FAST score and the histological data. However, we do believe that this is a brilliant study idea, which would be our next study topic. We really appreciate this comment once again.

  1. Did the authors find similar changes among other aromatic amino acids and FAST score like that of tyrosine?

Response: We appreciate this relevant comment. Alterations in circulating aromatic amino acids have been reported in patients with NAFLD, as was also mentioned in the Discussion, However, significant correlation was not found between higher FAST score and the other aromatic amino acids in our results as follows: phenylalanine, odds ratio (OR)=1.228 (95% confidence interval [CI], 0.960–1.571; P=0.102); tryptophan, OR=1.114 (95% CI, 95% CI, 0.875 – 1.419; P=0.381).

Reviewer 2 Report

Summary: The investigators prospectively collected 266 patients with available metabolomics and genotyping data. A total of 160 patients had follow-up studies during a median of 23.7 months.  Among them, 30 patients with weight loss >5% (18.7%), 75 patients with weight loss ≤5 (46.9%), and 55 patients with weight gain (34.4%). Using last FAST (cutoff ≤0.35) as outcome, they found that baseline older age, higher ALT, and higher tyrosine were significantly associated with risk toward a higher last FAST score.

Comments

1. If this is a prospective study, why only 266 of 423 patients with NAFLD had adequate data?  Follow-up timing and end point seems not prospectively arranged.

2. The FAST score was used as the outcome variable. Body weight changes were not a prognostic factor in model 2. Those weight gain groups also showed improvement of FAST score. Were there any metabolomics changes at the last follow-up? Such as muscular mass.

3. What will happen if weight changes serve as an outcome variate?

4. Were there any morbidity or mortality that occurred during the study period? Including those lost to follow-up.  

5. Outcome parameters of NAFLD, such as genetic polymorphism, may a need longer follow-up period to validate its value.  

Author Response

Summary: The investigators prospectively collected 266 patients with available metabolomics and genotyping data. A total of 160 patients had follow-up studies during a median of 23.7 months.  Among them, 30 patients with weight loss >5% (18.7%), 75 patients with weight loss ≤5 (46.9%), and 55 patients with weight gain (34.4%). Using last FAST (cutoff ≤0.35) as outcome, they found that baseline older age, higher ALT, and higher tyrosine were significantly associated with risk toward a higher last FAST score.

Comments

  1. If this is a prospective study, why only 266 of 423 patients with NAFLD had adequate data?  Follow-up timing and end point seems not prospectively arranged.

Response: We thank the reviewer for this important comment. During the study period, a total of 423 patients were enrolled for our NAFLD prospective cohort. Of these, however, blood samples for circulating metabolomics and genotyping were available in 266 patients. Therefore, a retrospective analysis was conducted utilizing data and samples from a prospectively enrolled cohort in the present study. We revised the Methods accordingly.

page 2, [2. Methods] 2.1 Study participants (2nd paragraph)

This was a retrospective analysis from our prospectively enrolled cohort.

  1. The FAST score was used as the outcome variable. Body weight changes were not a prognostic factor in model 2. Those weight gain groups also showed improvement of FAST score. Were there any metabolomics changes at the last follow-up? Such as muscular mass.

Response: We thank the reviewer for this important comment. The absence of significant effect of weight change on the outcome (Table 5) underscores the relevance of the significant factors including tyrosine, which showed independently significant association with the downward shift of the FAST score during follow-up, regardless of baseline weight or weight changes during follow-up. Regrettably, we were not able to evaluate potential effect of changes in circulating metabolites at follow-up due to lack of paired samples for metabolomics.

Despite the consistent relationship between muscle mass and the severity of NAFLD in previous studies (J Hepatol 2017;66:123-131, J Hepatol 2022;76:1021–1029), the absence of significance of sarcopenia on the outcome was an unexpected finding. We speculate that this finding might result from relatively low proportion of high-risk (FAST ≥0.67) patients (10.1%), suggesting less severe disease in the majority of the study participants. We revised the Discussion accordingly.

[Discussion, page 3, line 186]

In addition, sarcopenia was not significantly associated with the outcome, unlike previous reports [J Hepatol 2017;66:123-131, J Hepatol 2022;76:1021–1029], possibly because of relatively low proportion of high-risk (FAST ≥0.67) patients (10.1%).

  1. What will happen if weight changes serve as an outcome variate?

Response: We thank the reviewer for this constructive comment, given that FAST score at follow-up tended to be related to weight change as was depicted in Fig.1. We reiterated the multiple logistic regression analysis to identify potential predictive factors for weight change, with weight loss <5% as the reference. As shown in the table below, however, we were not able to find statistically significant variables related to weight change. This finding seems to be in line with our Results (Table 5), in which the relevant predictors for follow-up FAST score were significant regardless of baseline weight or weight changes.

Table. Multiple logistic regression analysis on risk factors for weight change

Weight gain

Weight loss ≥5%

OR

95% CI

P

OR

95% CI

P

PC ae C40:6

0.65

0.39-1.08

0.099

0.90

0.50-1.62

0.715

lysoPC a C18:2

0.86

0.54-1.37

0.525

0.83

0.45-1.54

0.555

SM C24:0

1.22

0.79-1.88

0.366

1.21

0.72-2.03

0.471

Tyrosine

0.98

0.60-1.60

0.927

0.56

0.29-1.09

0.087

Sex

1.59

0.59-4.3

0.357

1.10

0.31-3.84

0.886

Age

1.00

0.97-1.03

0.798

0.99

0.95-1.03

0.673

ALT

1.00

0.99-1.01

0.525

1.00

0.99-1.01

0.941

HOMA–IR

0.96

0.39-2.35

0.932

0.74

0.24-2.25

0.596

Sarcopenia

0.43

0.17-1.08

0.072

1.15

0.41-3.24

0.796

PNPLA3

0.38

0.15-0.97

0.044

0.58

0.18-1.80

0.342

Baseline weight

0.99

0.95-1.03

0.591

1.02

0.98-1.07

0.334

Abbreviations: OR, odds ratio; CI, confidence interval; SM, sphingomyeline; PC, phosphatidylcholine; lysoPC, lysophosphatidylcholine; ALT, alanine aminotransferase; HOMA-IR, homeostasis model assessment of insulin resistance

  1. Were there any morbidity or mortality that occurred during the study period? Including those lost to follow-up.  

Response: We appreciate this thoughtful comment. There were no serious morbidity/mortality captured during the study period (median follow-up period=23.7 months). However, because 106 participants were lost to follow-up, we included the remaining 160 patients with paired clinical evaluation to comprise the follow-up sub-cohort.

  1. Outcome parameters of NAFLD, such as genetic polymorphism, may a need longer follow-up period to validate its value.  

Response: We appreciated this valuable comment. Based on our results, we will design a subsequent study with longer follow-up, including more institutions. We thank the reviewer for this idea once again.